# Resistance and Biofilm Production Profile of Potential Isolated from *Kpètè-Kpètè* Used to Produce Traditional Fermented Beer

**DOI:** 10.3390/microorganisms11081939

**Published:** 2023-07-29

**Authors:** Christine N’Tcha, Haziz Sina, Dyana Ndiade Bourobou, S. M. Ismaël Hoteyi, Bawa Boya, Raoul Agnimonhan, Jacques François Mavoungou, Adolphe Adjanohoun, Olubukola Oluranti Babalola, Lamine Baba-Moussa

**Affiliations:** 1Laboratory of Biology and Molecular Typing in Microbiology, Department of Biochemistry and Cell Biology, University of Abomey-Calavi, Abomey-Calavi 05 BP 1604, Benin; nekis90@yahoo.fr (C.N.); sina.haziz@gmail.com (H.S.); smihot@yahoo.fr (S.M.I.H.); boyabawa@gmail.com (B.B.); agnimonhanraoul@yahoo.fr (R.A.); 2Institut de Recherches Agronomiques et Forestières (IRAF), BP.12978 Gros-Bouquet, Libreville B.P. 16 182, Gabon; dyananbd@gmail.com (D.N.B.); mavoungoujacques@yahoo.fr (J.F.M.); 3National Agronomic Research Institute of Benin, Cotonou 01 BP 884, Benin; adjanohouna@yahoo.fr; 4Food Security and Safety Focus Area, Faculty of Natural and Agricultural Sciences, North-West University, Private Bag X2046, Mmabatho 2735, South Africa

**Keywords:** *kpètè-kpètè*, microbiological contaminant, *Enterobacteria*, resistance genes, *Staphylococcus* spp.

## Abstract

This study aimed to characterize the pathogenicity of bacteria isolated from the starter of two traditional beers produced and consumed in Benin. After standard microbial identification, species were identified by specific biochemical tests such as catalase, coagulase, and API 20 E. Antibiotic sensitivity was tested according to the French Society of Microbiology Antibiogram Committee. The crystal violet microplate technique evaluated the biofilm production and conventional PCR was used to identify genes encoding virulence and macrolide resistance. According to our data, the traditional starter known as *kpètè-kpètè* that is used to produce beer is contaminated by *Enterobacteriaceae* and staphylococci species. Thus, 28.43% of the isolated bacteria were coagulase-negative staphylococci (CNS), and 10.93% coagulase-positive staphylococci (CPS). Six species such as *Klebsiella terrigena* (1.38%), *Enterobacter aerogens* (4.14%)*, Providencia rettgeri* (5.51%)*, Chryseomonas luteola* (6.89%)*, Serratia rubidae* (15.16%), and *Enterobacter cloacae* (27.56%) were identified among *Enterobacteriaceae*. Those bacterial strains are multi-resistant to conventional antibiotics. The hight capability of produced biofilms was recorded with *Enterobacter aerogens*, *Klebsiella terrigena* (100%), *Providencia rettgeri* (75%), and *Staphylococcus* spp (60%). *Enterobacter cloacae* (4%) and coagulase-negative *Staphylococcus* (5.55%) harbor the macrolide resistance gene. For other strains, these genes were not detected. Foods contaminated with bacteria resistant to antibiotics and carrying a virulence gene could constitute a potential public health problem. There is a need to increase awareness campaigns on hygiene rules in preparing and selling these traditional beers.

## 1. Introduction

Food poisoning is a serious issue that can cause more than 200 different illnesses, ranging from diarrhea to cancer, and results in nearly 420,000 deaths each year worldwide [1]. Unfortunately, street food is often a culprit when it comes to incidents of food poisoning [2]. This is particularly true in Benin, where *Tchoukoutou* and *Tchapalo*, two traditional beers made from red or brown sorghum (*Sorghum bicolor*) and maize [3,4], are sold on the street. These beers are an important part of social gatherings and traditional ceremonies [5] and are a significant income source, particularly for women [3,6]. However, there have been reports of a lack of hygiene during the cooking and sale of these beers, which can lead to contamination with pathogenic microorganisms that cause food poisoning. 

The production of *Tchoukoutou* and *Tchapalo* involves fermentation with microorganisms that create various microbial communities. These communities significantly impact the final products’ sensory quality, nutrient availability, and safety [7,8,9]. Bacteria, yeasts, and molds can be found in the raw materials, utensils, or even the producers themselves [10,11]. Although lactic acid bacteria can reduce the levels of pathogenic bacteria present in the drinks [7,8,12,13], some bacteria such as *Enterobacter sakazaki*, *Klebsiella pneumonia*, *Escherichia coli*, and *Staphylococcus aureus* remain resistant to the acidic environment and cooking process of the fermented foods [14]. *Enterobacteriaceae* in *Tchoukoutou*, sold in northern Benin, has been reported [15]. It is concerning that the bacteria found in beer can lead to food spoilage [16,17] and negatively impact individuals with health conditions or habitat disorders, such as sepsis, meningitis, endocarditis, peritonitis, or heart disease [18]. 

In West African nations, the prevalence of bacterial infections has risen alongside the increase in antibiotic consumption [19,20]. Improper and unregulated use of synthetic products has made it difficult to control bacterial and fungal infections, as many conventional antibiotics and antifungals have become ineffective due to the emergence of resistant bacteria and fungi [21,22]. This resistance to multiple drugs may be caused by chromosomal mutations in these strains or by acquiring resistance genes [23]. However, the *Enterobacteriaceae* species and many pathogenic bacteria found in traditional beers have been overlooked in the literature, despite being identified through biochemical and molecular characterization. Thus, the present study aims to characterize the pathogenic bacteria isolated from two traditional beers produced and consumed in Benin. 

## 2. Materials and Methods

### 2.1. Sampling

Thirty-seven (37) isolates of the enterobacteria and staphylococci used were isolated from samples of *kpètè-kpètè* of two traditional beers (*Tchoukoutou* and *Tchapalo*) collected by N’tcha et al. [15] in areas with a high production of conventional beers in Benin such as Tanguiéta, Natitingou, Parakou, Boukoumbé, Tchaourou, N’Dali, Bantè, Dassa Zoumè, and Glazoué. 

### 2.2. Microbiological Analyses

#### 2.2.1. Identification of Bacterial Strains

In the laboratory, 5 mL of Mueller Hinton broth was added to each sample. The cases were incubated at 37 °C for 24 h. Following incubation, areas with cloudy points indicating bacterial growth were selected for germ testing. Isolation and purification of strains were carried out using selective media. Mac Conkey, EMB, and SS Agar media were used to isolate *Enterobacteriaceae*, and Baird Parker medium supplied with egg yolk + potassium tellurite + 0.2% sulfamethazine and a coagulase test were used for the staphylococci. To ensure the purity of the sample, the isolated colony was subcultured for isolation. The pure strains were obtained after three successive subcultures were used for biochemical and molecular identification. We utilized the API 20E commercial kit from BioMerieux (Marcy l’Etoile, France) to identify isolated *Enterobacteriaceae*. The identification process was done through plating. 

#### 2.2.2. Biofilm Formation

The ability of the isolates to generate a biofilm was assessed through a process described in reference [24]. To begin, 10 µL of 18-h-old isolate suspension was diluted with 150 µL of Brain Heath Infusion, and the mixture was incubated for 24 h at 37 °C. Once incubated, wells were washed three times with approximately 0.2 mL of sterile physiological water containing 0.9% NaCl. To observe the biofilms created by the attachment of stationary organisms to the microplate in each well, crystal violet (0.1%) was used to stain them for 10 min. The violet staining in a well indicated a positive test. The plates were washed thoroughly to remove excess dye and left to dry at room temperature [25].

#### 2.2.3. Susceptibility of Bacterial Isolates to Tested Antibiotic 

To assess the antibiotic susceptibility of the isolated strains, we utilized the disc diffusion method, per the guidelines set forth by the Antibiogram Committee of the French Society of Microbiology [26]. For the *Enterobacteria*, we utilized a range of antibiotics from Oxoid^®^, including ampicillin (A, 10 µg), cefoxitin (FOX 30 µg), ceftriaxone (Cl, 30 µg), cefotaxime (CTX 30 µg), chloramphenicol (C 30), gentamicin (G 10 µg), doxycycline (DO 30 µg), nalidixic acid (NA 30 µg), and trimethoprim-sulfamethoxazole (STX 25 µg). The antibiotics agents (Oxoid^®^) used for staphylococci were penicillin G (1 µg), ceftriaxone (Cl 30 µg), cefotaxime (CTX 30 µg), cephalothin (KF 30 µg), vancomycin (VA 5µg), chloramphenicol (C 30), gentamicin (G 10 µg), streptomycin (S 10 µg), erythromycin (E 15 µg), trimethoprim-sulfamethoxazole (STX 25 µg), and nalidixic acid (NA 30 µg).

### 2.3. Molecular Characterization of Isolates

#### 2.3.1. DNA Extraction

DNA extraction was performed according to an adaptation of the previously described method by Rasmussen et Morrissey [27]. To prepare for DNA extraction, 3–4 colonies from a fresh bacterial culture (18 h old) were used for preculture in 1 mL of brain heart infusion and incubated at 37 °C for 18 h. The resulting mixture was centrifuged at 12,000 rpm for 5 min, and the supernatant was discarded. The bacterial pellet was mixed with 500 µL of TBE 1x and heated at 95 °C for 15 min in a dry bath. After centrifuging the mixture at 12,000 rpm for 5 min, the supernatant was transferred to a sterile tube and mixed with 500 µL of ethanol. The DNA pellets were then suspended in 50 µL of distilled water and stored at +4 °C for immediate use or at −20 °C for long-term storage.

#### 2.3.2. Search for Alleles of Genes Encoding Virulence

For *Enterobacteriaceae*, genes encoding for the synthesis of adhesins (*fimA: fim* AF*:* 5′-CGGCTC TGTCCCTSAGT-3′ and *fim* AR: 5′-GTCGCATCCGCATTAGC-3′) [28,29] and the one encoding for the synthesis of the cytotoxic necrosis factor (*cnf1*: *cnf1F: 5′*-GGGGGAAGTACAG AAGAATTA-3′ and *cnf1R:* 5′-TTGCCGTCCACTCTCACCAGT-3′) [30,31] were searched for. The reaction mix included 20 µL of 10x GoTaq (PROMEGA, Madison, WI, USA) master mix, 1 µL of 10 µM primer F, 1 µL of 10 µM primer R, and 3 µL of DNA extract. The PCR reactions were performed using the thermal cycler using the following program: a cycle of initial denaturation (94 °C/3 min), followed by 30 cycles of denaturation (94 °C/1 min), hybridization (52 °C/1 min), and elongation (72 °C/1 min). To end, a final extension (72 °C/10 min) was performed.

Concerning the *Staphylococcus* spp isolates, gene alleles encoding macrolide resistance (*ermB*
*ermB1: 5′-*FGAAAAGGTACTCAACCAAATA-3′ and *ermB2:* 5′-AGTAACGGTAC TTAAATTGTTTAC-3′) [32] and gene alleles encoding erythromycin resistance (*mefA*: *mef(A/E)*1*:* 5′-AGTATCATTAATCACTAGTGC-3′ and *mef(A/E)*2*:* 5′-TTCTTCTGGTAC TAAAAGTGG-3′) [32] were investigated. The reaction mix included 20 µL of 10x GoTaq (PROMEGA, Madison, WI, USA) master mix, 1 µL of 10 µM primer F, 1 µL of 10 µM primer R, and 3 µL of DNA extract. The amplification process involved an initial denaturation at 95 °C for 3 min. This was followed by 35 cycles of denaturation, taking 1 min at 95 °C, hybridization for 1 min at 57 °C, and elongation for 1 min at 72 °C. Finally, a final extension was carried out for 10 min at 72 °C. 

### 2.4. Data Analysis and Processing

The results of all experiments carried out in duplicate were plotted on a bench sheet and then entered into the Excel 2016 spreadsheet. The Excel API 20 E spreadsheet was used to identify the biochemical profiles of the strains. The data were subjected to an analysis of variance (ANOVA) using the GraphPad Prism 8 software. Tukey’s multiple comparisons test was used to compare the difference in means. A statistically significant value was determined to be when *p* < 0.05. 

## 3. Results

### 3.1. Identification of Pathogenic Bacteria Isolated

Ninety-four bacteria strains containing *Enterobacteriaceae* (60.64%) and staphylococci (39.36%) were isolated. Of the isolated bacteria, 28.43% were coagulase-negative staphylococci (CNS) and 10.93% were coagulase-positive staphylococci (CPS). Concerning the *Enterobacteriaceae*, six different species such as *Klebsiella terrigena* (1.38%), *Enterobacter aerogens* (4.14%), *Providencia rettgeri* (5.51%), *Chryseomonas luteola* (6.89%), *Serratia rubidae* (15.16%), and *Enterobacter cloacae* (27.56%) were isolated. However, the most predominant species remained CNS and *E. cloacae* (Figure 1).

### 3.2. Biofilm Production

Among the isolated *Enterobacteriaceae*, 44% were biofilm producers. The biofilm production rate significantly varied (*p* < 0.0001) according to the species (Figure 2). Thus, Figure 2 shows that 100% *of Enterobacter aerogens* and *Klebsiella terrigena* were biofilm producers, following *Providencia rettgeri* (75%). The lowest production rate was observed with *Serratia rubidaea* (14.28%). Concerning the staphylococci, 28% were biofilm producers. As shown in Figure 2, this production level was high among coagulase-positive staphylococci (60%) compared to coagulase-negative staphylococci (15.38%).

### 3.3. Resistance Profile of Enterobacteriaceae to Antibiotics by Isolated Species

The isolated *Enterobacteriaceae* were divided into 6 species to evaluate their antibiotic susceptibility. Thus, Table 1 shows that *Serratia rubidae*, *Enterobacter cloacae*, *Enterobacter aerogens*, *Providencia rettgeri*, *Chryseomonas luteola*, and *Klebsiella terrigena* were highly resistant (100%) to ampicillin, chloramphenicol, cefoxitin, and ceftriaxone. These strains showed sensitivity to gentamicin (1.85%), doxycycline (32.4%), trimethoprim/sulfamethoxazole (32.87%), and nalidixic acid (43.51%). Nevertheless, *Klebsiella terrigena* and *Enterobacter aerogens* were 100% resistant to quinolones such as nalidixic acid and developed, respectively, high immunity to cyclones such as doxycycline and sulfonamides such as trimethoprim-sulfamethoxazole. The statistical analysis of Table 1 shows a significant difference between the effect of using antibiotics (*p* < 0.001).

Gentamicin completely inhibited all strains of *Enterobacteriaceae* without *Enterobacter cloacae*, but only 11.11% of *Enterobacter cloacae* showed resistance to this antibiotic. Figure 3 shows that only *Enterobacter cloacae* developed resistance to all antibiotics, followed by *Providencia rettgeri*, while *Chryseomonas luteola* remained the most sensitive strain. 

### 3.4. Resistance Profile of Staphylococci to Antibiotics by Species

Staphylococci were divided into two categories for assessing susceptibility to antibiotics: *Staphylococcus aureus* (CPS) and *Staphylococcus* spp (CNS). Figure 4 shows the results of the antibiotic resistance profile of staphylococci.

Based on the statistical analysis of Table 2, it is evident that there existed a significant disparity in the usage of antibiotics (*p* < 0.001). Table 2 shows that all strains of staphylococci were 100% resistant to β-lactams, particularly penicillin G, cefoxitin, cephalothin, chloramphenicol, and ceftriaxone, sulfonamides such as trimethoprim-sulfamethoxazole, glycopeptides such as vancomycin, and macrolides such as erythromycin. However, their resistance was low to gentamicin (31.54%), nalidixic acid (39.23%), and streptomycin (59.92%). Gentamicin showed a low efficacy against staphylococci, particularly *Staphylococcus aureus.*

### 3.5. Virulence Genes and Resistance Gene Macrolides

In this study, 4% of *Enterobacteriaceae* strains harbored *the fimA* gene, and 5.55% of staphylococci strains contained the *ermB* gene encoding macrolide resistance (Table 3). However, there is no *cnf1* and *mefA* gene for *Enterobacteriaceae* exhibiting enterotoxin produced by *E. coli* and staphylococci showing erythromycin resistance.

## 4. Discussion

The primary source of human infections is ingesting contaminated food [33]. Indeed, pathogenic bacteria such as staphylococci and *Enterobacteriaceae* have been identified in African fermented cereal foods [34] and traditional fermented drinks [15]. Our study showed that the percentages of samples contaminated with these microorganisms were 60.64% of *Enterobacteriaceae* and 39.36% of staphylococci. This rate of *Enterobacteriaceae* contamination is much higher than the 14.4% obtained by Cason et al. [35] in Sesotho, a traditional beer from South Africa produced from corn or sorghum. The low rate of *Enterobacteriaceae* and the absence of staphylococci in Sesotho can be explained by the fact that Sesotho is obtained from two fermentations with the addition of three different ferments (Tomoso, Mmela, and Yomoso) and by the fact that the two studies were not carried out under the same conditions. However, observing the rules of Good Hygiene Practices and Good Preparation Practices can reduce contamination rates in this situation. 

The biochemical characterization of the enterobacteria and staphylococci isolated strains gave us more precision as to the species in these traditional beers. We noted the presence of *Klebsiella terrigena* (2.28%), *Enterobacter aerogens* (6.82%), *Providencia rettgeri* (9.09%), *Chryseomonas luteola* (11.36%), *Serratia rubidae* (25%), and *Enterobacter cloacae* (45.45%) for *Enterobacteriaceae*. As for the *Staphylococcus* genus, we noted the presence of 13.51% of coagulase-positive staphylococci (probably *Staphylococcus aureus*) and 86.48% of coagulase-negative staphylococci (*Staphylococcus* spp). Certain pathogenic microorganisms and alterations of traditional beers observed in Sesotho, namely, the genus’ *Chryseobacterium*, *Enterobacter*, and *Klebsiella* [35], were also present in our samples. These strains have also been isolated in other traditional beers prepared from raw materials, such as rice for Chicha, a Brazilian fermented drink [36], and cactus juice for pulque, a Mexican alcoholic beverage [37]. The high proportion of *Enterobacter cloacae* (45.45%) observed in these traditional beers could be due to their habitat, which remains the environment. The results of our study corroborate those of several studies [35,36,37] by highlighting the presence of unusual species such as *Serratia rubidae*, *Providencia rettgeri*, and *Staphylococcus aureus* in our samples. Indeed, *Serratia rubidae* and *Providencia rettgeri* have been found in stool [38] and flies [39], respectively. Staphylococci are ubiquitous natural bacteria, i.e., they are found in air, water, soil, food, furniture, and equipment [18,40]. The presence of these different species can be explained by their contaminating of the beer production process, raw materials, the sales environment, lack of supervision, or non-compliance with the hygiene rules. 

During bacterial infections, the way of life developed by bacteria is biofilm production [41]. The production of biofilm is one of the defense mechanisms of bacteria. Biofilm formation is a default lifestyle for staphylococci and *Enterobacteriaceae* [42,43]. The high rate of biofilm production observed in *Enterobacter aerogens* and *Klebsiella terrigena* during our research can be explained by the fact that they are Gram-negative thermotolerant bacilli of food origin. Regarding staphylococci, the low rate observed in coagulase-negative Staphylococci (15.35%) was also found in the work of Ahouandjinou et al. [44], with a respective proportion of biofilm production of 28% for *Staphylococcus lugdunensis* and 20% for *S. warneri* on food coagulase-negative *Staphylococcus* (CNS) strains. Indeed, some CNSs, such as *Staphylococcus epidermidis*, adhere more quickly to the stainless steels of food industry equipment [45]. Therefore, these CNSs can easily comply with the kitchen utensils involved in producing these traditional beers. Therefore, properly cleaning these equipment and kitchen utensils could justify the low rate observed in the CNS. As for *S. aureus* (60%), the high rate observed can be explained by the fact that they adhere and quickly develop biofilm in contact with the food surface [46]. This lifestyle facilitates the adhesion of bacteria to food surfaces creating a public health problem [47]. In short, in these beers, the high rate of contamination observed with *Providencia rettgeri*, *Enterobacter aerogens*, *Klebsiella terrigena*, and *Staphylococcus aureus* can be explained by the fact that they can produce a biofilm, which provides specific adhesion characteristics allowing for their persistence and their adaptation to bad conditions. The production of biofilms is potentially a risk of poisoning for the consumer. 

Gentamicin and nalidixic acid had shown efficacy in almost all strains isolated except for *Enterobacter cloacae*. Our results corroborate those of Hama et al. [48] and Anago et al. [49], who qualify gentamicin as having excellent efficacy against staphylococci and ESBL-producing strains of enterobacteria, respectively. This effectiveness of gentamicin, accompanied by nalidixic acid, can be explained by the fact that aminoglycosides (gentamicin) act on the bacterial ribosome and induce the synthesis of erroneous proteins [50]. Regarding quinolones, such as nalidixic acid, their mechanism of action involves targeting DNA gyrase and topoisomerase IV during DNA replication. These enzymes regulate the DNA topology necessary for replication [51]. The inhibition of DNA gyrase leads to the suppression of the positive supertowers of DNA.

In contrast, that of topoisomerase IV leads to the accumulation of daughter replicons, thus disrupting this replication. In addition to gentamicin and nalidixic acid, we note that some antibiotics, such as doxycycline and trimethoprim-sulfamethoxazole, remain active on enterobacteria, as does streptomycin on staphylococci. A study conducted by Kadja et al. [52] on combining tetracycline and trimethoprim sulfonylurea on strains from dairy farming corroborated our results with an efficacy of 62% and 67%, respectively. As for β-lactams and others, the *Enterobacteriaceae* were all resistant (100%). Previous work corroborates our results on the sensitivity of *Enterobacteriaceae* strains isolated to the antibiotics tested [53,54]. In addition, some work has shown that vancomycin, penicillin G, and macrolides are effective against food-borne *S. aureus* strains [52,55]. Globally, the resistance of *Enterobacteriaceae* and staphylococci isolates to β-lactams can be explained by the production of β-lactamase, which hydrolyzes these antibiotics or the structure of the pores is reduced by the antibiotics that pass [56]. The development of resistance to other antibiotics can be explained by the fact that these strains have already been in contact with different families of antibiotics. Particularly for staphylococci, resistance to ceftriaxone implies resistance to almost all β-lactams currently available [57] and the development of resistance to many antibiotics widely used to control infections such as food poisoning [58,59]. The multidrug resistance of the isolates could be explained by self-medication and the excessive and uncontrolled use of antibiotics accompanied by a selection of resistant bacteria [60,61]. This selection could be due to acquiring resistance genes such as *erm*, *msr*, *mef*, and *mph* [56], or virulence genes such as *fimA* and *cnf1* [62]. 

Our study only revealed a shallow presence of *fimA* and ermB genes in *Enterobacteriaceae* (*Enterobacter cloacae*) and staphylococci (CNS). The work of Le Trong [63] showed the *fimA* gene’s presence in many *Enterobacteriaceae* species. In 2020, Soria-Bustos et al. [64] isolated *Enterobacter cloacae* with virulence properties in the appearance of extra-intestinal infections. Thus, the biofilm production capacity of *E. cloacae* could be explained by the *fimA* gene, which codes for the synthesis of adhesin, an essential protein in biofilm formation. As for the *cnf1* gene, its absence can be explained by the fact that it is mainly produced by *E. coli* [65]. Regarding staphylococci, a study on the transferability of antibiotic resistance genes between bacteria carried out during the food fermentation of milk showed that *Staphylococcus aureus* and potentially pathogenic coagulase-negative staphylococci are carriers of macrolide resistance genes [66,67]. 

However, the low proportion of *ermB* genes with a complete absence of *mefA* genes, even though staphylococci were phenotypically resistant to erythromycin in our study, can be explained by the fact that these staphylococci could develop other resistance mechanisms through the acquisition of other genes such as *msr*, *mph,* and *ere*, encoding for macrolides resistance. Apart from the addition of genes, the multidrug resistance of strains can be explained by forming a biofilm that protects bacteria from antibacterial molecules [68]. There is a correlation between the ability to form biofilms and antibiotic resistance. This antibiotic resistance can be explained by the diffusion of drugs in the biofilm [69]. After exposure to these agents, a small surviving population can repopulate the surface immediately and become more resistant to antimicrobial treatment. Food contaminated with bacteria resistant to antibiotics and carrying genes for resistance or virulence are dangerous germs that could threaten public health. 

We must acknowledge the limitations of our study, as we solely analyzed samples from the northern region of the country. It would be beneficial to expand our sampling to encompass other areas. Additionally, we could broaden our investigation to include other pathogenic strains and examine further pathological factors. 

## 5. Conclusions

From our research, it was found that the *kpètè-kpètè* used to produce traditional fermented beers (*Tchoukoutou* and *Tchapalo*) are contaminated with pathogenic bacteria such as *Serratia rubidae*, *Enterobacter cloacae*, *Enterobacter aerogens*, *Providencia rettgeri*, *Chryseomonas luteola*, *Klebsiella terrigena*, and *Staphylococcus aureus*. These bacteria exhibit multidrug resistance to antibiotics with the ability to form biofilms. The presence of these strains in traditional drinks is a reality and remains worrying for the consumer’s health. It would be better to extend this study to other types of fermented foods, to deepen this study on the search for bacterial toxins and genes for antibiotic resistance given the phenotypic character resistant to macrolides in general and erythromycin that we observed in this study.

## Figures and Tables

**Figure 1 microorganisms-11-01939-f001:**
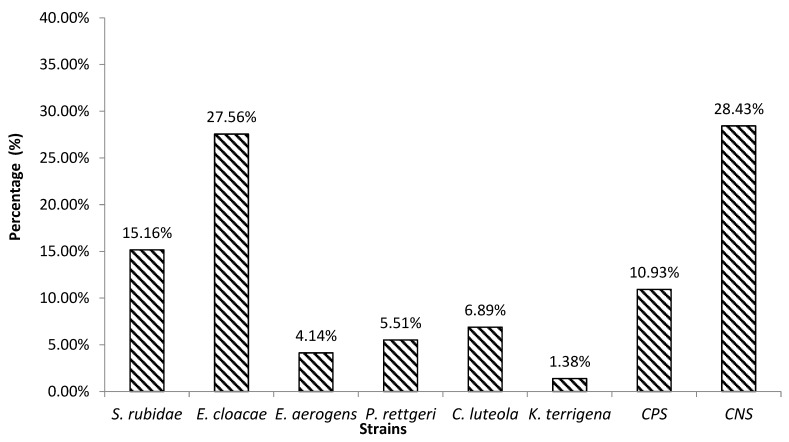
Distribution of *Enterobacteriaceae* and staphylococci strains isolated from *kpètè-kpètè* samples.

**Figure 2 microorganisms-11-01939-f002:**
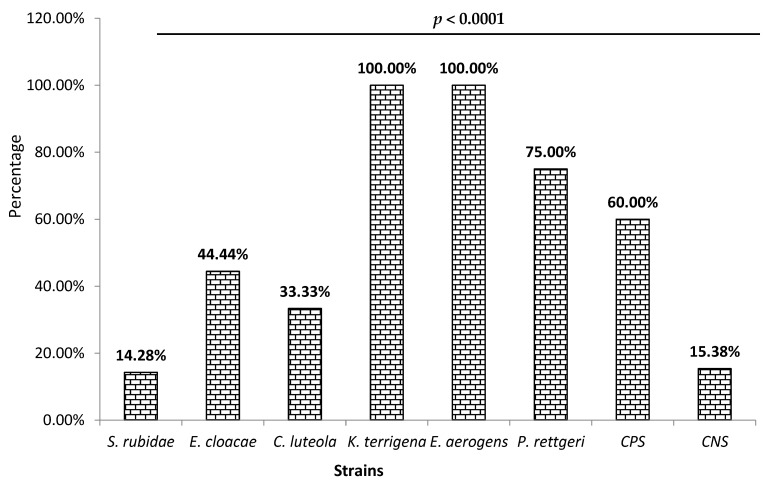
Distribution of biofilm production capacity according to identified species.

**Figure 3 microorganisms-11-01939-f003:**
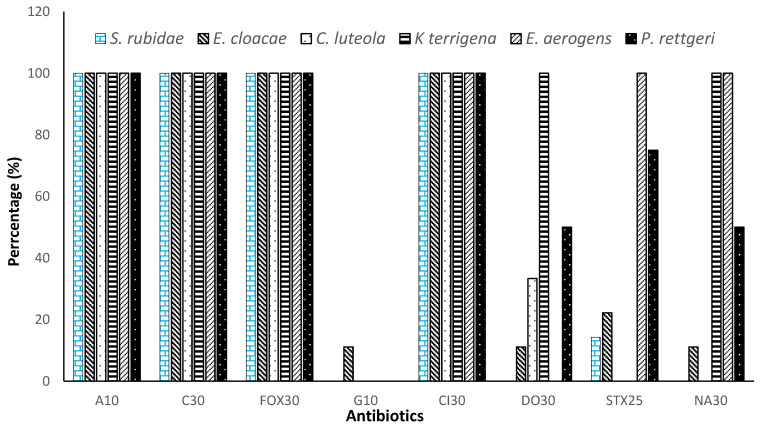
Resistance profile of isolated *Enterobacteriaceae* to tested antibiotics. A10: Ampicillin (10 μg), C30: Chloramphenicol (30 µg), FOX30: Cefoxitin (30 µg), G10: Gentamicin (10 µg), Cl30: ceftriaxone (30 µg), DO30: Doxycycline (30 µg), SXT25: trimethoprim -sulfamethoxazole (25 µg), NA30: nalidixic acid (30 µg).

**Figure 4 microorganisms-11-01939-f004:**
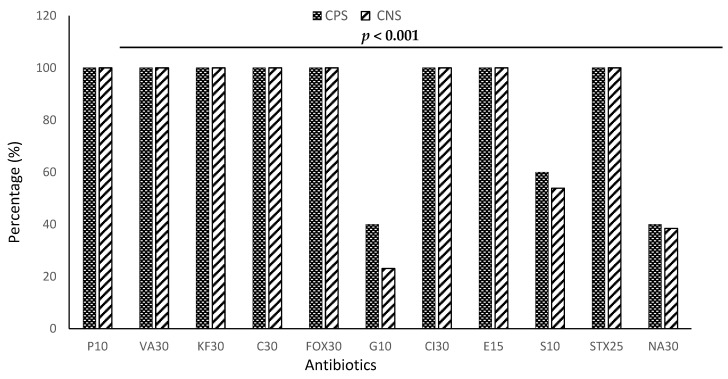
Resistance profile of staphylococci to tested antibiotics. CPS: *Staphylococcus aureus*, CNS: *Staphylococcus* spp, P 10: Penicillin (10 μg), VA30: Vancomycin (30 μg), KF 30: Cephalothin (30 μg), C30: Chloramphenicol (30 µg), FOX: Cefoxitin (30 µg), G10: Gentamicin (10 µg), Cl30: ceftriaxone (30 µg), E15: Erythromycin(15 μg), S10: Streptomycin (10 μg), SXT2: trimethoprim-sulfamethoxazole (25 µg), NA30: Nalidixic acid (30 µg).

**Table 1 microorganisms-11-01939-t001:** Distribution of *Enterobacteriaceae* resistant phenotypes to antibiotics.

Antibiotics	Percentage of Resistance	*p* Value
Ampicillin (A10)	100.00%	**<0.001**
Ceftriaxone (Cl30)	100.00%	
Cefoxitin (FOX30)	100.00%	
Chloramphenicol (C30)	100.00%	
Gentamicin (G10)	1.85%	
Doxycycline (DO30)	32.40%	
Trimethoprim-sulfamethoxazole (STX25)	32.85%	
Nalidixic acid (NA30)	43.51%	

**Table 2 microorganisms-11-01939-t002:** Distribution of antibiotics-resistant phenotypes of staphylococci.

Antibiotics	Percentage of Resistance	*p* Value
Penicillin G (P10)	100.00%	**<0.0001**
Ceftriaxone (Cl30)	100.00%	
Cefoxitin (FOX30)	100.00%	
Chloramphenicol (C30)	100.00%	
Cephalothin (KF30)	100.00%	
Vancomycin (VA30)	100.00%	
Erythromycin (E15)	100.00%	
Trimethoprim-sulfamethoxazole (STX25)	100.00%	
Gentamicin (G10)	31.54%	
Nalidixic Acid (NA30)	39.23%	
Streptomycin (S10)	59.92%	

**Table 3 microorganisms-11-01939-t003:** Distribution of virulence and resistance genes.

	*fimA*	*cnf1*	*ermB*	*mefA*
*Klebsiella terrigena*	(0%)	(0%)	*NA*	*NA*
*Enterobacter aerogens*	(0%)	(0%)	*NA*	*NA*
*Providencia rettgeri*	(0%)	(0%)	*NA*	*NA*
*Chryseomonas luteola*	(0%)	(0%)	*NA*	*NA*
*Serratia rubidae*	(0%)	(0%)	*NA*	*NA*
*Enterobacter cloacae*	(4%)	(0%)	*NA*	*NA*
*S. aureus*	*NA*	*NA*	(0%)	(0%)
*SCN*	*NA*	*NA*	(5.55%)	(0%)

*NA*: Not applicable.

## Data Availability

The data used to support the findings of this work are available from the corresponding author upon request.

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
