# Peer review of "Resistance and Biofilm Production Profile of Potential Isolated from Kpètè-Kpètè Used to Produce Traditional Fermented Beer"

_microorganisms, 2023, doi:10.3390/microorganisms11081939_

Round 1

Reviewer 1 Report

In this study, the dominant bacteria Enterobacteriaceae spp. and Staphylococcus spp. were identified and isolated from the kpètè-kpètè of traditional brewing beer. The author analyzed their biofilm formation ability, antibiotic resistance, and toxicological genes. It is intended to improve the understanding of food safety and hygiene rules through the study of strains isolated during the manufacture of traditional beer. The language of this article is fluent, and the overall architecture is clear. However, the article is not rigorous in many expressions and formats. The issues listed below require more explanation or revision.

1. The title of Figure 1 in line 158 needs to be changed because there are not only Enterobacteriaceae strains but also Staphylococci.

2. “Enterobacteriaceae” in line 161 needs italics.

3. Since the meanings of CPS and CNS have been explained in the text, in order to be consistent with Figure 1, it is suggested to delete the interpretation of them in line 170.

4. “Percentage of resistance” in Table 2 is not shown due to format problems; the merger is recommended in p> 0.9999 of different antibiotics; In addition, if the analysis of the whole table P < 0.001, it is recommended to add an additional column to distinguish the front.

5. The title of line 194 is wrong. It should be changed into “Resistance profile of Staphylococci to antibiotics by species”.

6. There are no relevant data results presented in result 3.5.

7. The analysis of the characteristics of the isolated dominant bacterial groups is missing from the discussion part.

8. There is a difference between the format 1-9 and 10-69 in ref.

The language of this article is fluent, and the overall architecture is clear.

Reviewer 2 Report

The subject of the study is an interesting one, but some additions are necessary for publication:

1- Figures 1 and 2 inside the graphs, write the results correctly, with a period, not a comma.

2- It is necessary to include the statistical part of the results in the graphic representation.

3 - The limits of the study must also be mentioned.

Reviewer 3 Report

Although it is an interesting study, however, I found a major problem with the shallow depth of studies. Moreover, the manuscript is poorly written. There are a lot of problems in the manuscript particularly the structure/describing style and language of the manuscript. My individual comments are listed below:

The abstract needs a substantial revision. The aim of the study pointed in the abstract is not clear and does not represent well the essence of the study.

The introduction is not structured properly and the information presented is very messy.

Line 161 – The names of microorganisms should be in italics. 

Materials and Methods. It was difficult because the text is not understandable in some places. Please verify the method description. 

References- Please check the format of the references. I recommend checking the guide for reference format.

The number of bibliographic sources is adequate, but less than 20% of the total bibliographic sources are from the last 5 years, please supplement with more current bibliographic sources

There are some grammatical errors and instances of badly worded/constructed sentences throughout the manuscript. Please refine the language carefully. 

Round 2

Reviewer 2 Report

The way of expression must be checked carefully. Also the correct writing of the names of the microbial strains.

Reviewer 3 Report

The authors made several amendments to the manuscript following the provided suggestions. As such, the quality of the manuscript has been overall improved.

To my opinion, this revised manuscript can be accepted. 

Author Response

We would like to thank the reviewer for this endorsement.

Best regards